# Using Convolutional Neural Networks to Recognize Rhythm Stimuli from Electroencephalography Recordings

**Sebastian Stober, Daniel J. Cameron and Jessica A. Grahn**
Brain and Mind Institute, Department of Psychology, Western University
London, Ontario, Canada, N6A 5B7
{sstober,dcamer25,jgrahn}@uwo.ca

## Abstract

Electroencephalography (EEG) recordings of rhythm perception might contain enough information to distinguish different rhythm types/genres or even identify the rhythms themselves. We apply convolutional neural networks (CNNs) to analyze and classify EEG data recorded within a rhythm perception study in Kigali, Rwanda which comprises 12 East African and 12 Western rhythmic stimuli – each presented in a loop for 32 seconds to 13 participants. We investigate the impact of the data representation and the pre-processing steps for this classification tasks and compare different network structures. Using CNNs, we are able to recognize individual rhythms from the EEG with a mean classification accuracy of 24.4% (chance level 4.17%) over all subjects by looking at less than three seconds from a single channel. Aggregating predictions for multiple channels, a mean accuracy of up to 50% can be achieved for individual subjects.

## 1 Introduction

Musical rhythm occurs in all human societies and is related to many phenomena, such as the perception of a regular emphasis (i.e., beat), and the impulse to move one's body. It is a universal human phenomenon, but differs between human cultures. The influence of culture on the processing of rhythm in the brain as well as the brain mechanisms underlying musical rhythm are still not fully understood. In order to study these, we recruited participants in East Africa and North America to test their ability to perceive and produce rhythms derived from East African and Western music. Besides several behavioral tasks, which have already been discussed in [1], the East African participants also underwent electroencephalography (EEG) recording while listening to East African and Western musical rhythms thus enabling us to study the neural mechanisms underlying rhythm perception.

Using two popular deep learning techniques – stacked denoising autoencoders (SDAs) [2] and convolutional neural networks (CNNs) [3] – we already obtained encouraging early results for distinguishing East African and Western stimuli in a binary classification task based on the recorded EEG [4]. In this paper, we address the much harder classification problem of recognizing the 24 individual rhythms. In the following, we will review related work in Section 2, describe the data acquisition and pre-processing in Section 3, present our experimental findings in Section 4, and discuss further steps in Section 5.

## 2 Related work

How the brain responses to auditory rhythms has already been investigated in several studies using EEG and magnoencephalography (MEG): Oscillatory neural activity in the gamma (20-60 Hz) frequency band is sensitive to accented tones in a rhythmic sequence and anticipates isochronous tones [5]. Oscillations in the beta (20-30 Hz) band increase in anticipation of strong tones in a non-isochronous sequence [6, 7, 8].

Another approach has measured the magnitude of steady state evoked potentials (SSEPs) (reflecting neural oscillations entrained to the stimulus) while listening to rhythmic sequences [9, 10]. Here, enhancement of SSEPs was found for frequencies related to the metrical structure of the rhythm (e.g., the frequency of the beat). In contrast to these studies investigating the oscillatory activity in the brain, other studies have used EEG to investigate event-related potentials (ERPs) in responses to tones occurring in rhythmic sequences. This approach has been used to show distinct sensitivity to perturbations of the rhythmic pattern vs. the metrical structure in rhythmic sequences [11], and to suggest that similar responses persist even when attention is diverted away from the rhythmic stimulus [12]. Further, Will and Berg [13] observed a significant increase in brain wave synchronization after periodic auditory stimulation with drum sounds and clicks with repetition rates of 1–8Hz. Vlek et al. [14] already showed that imagined auditory accents can be recognized from EEG. They asked ten subjects to listen to and later imagine three simple metric patterns of two, three and four beats on top of a steady metronome click. Using logistic regression to classify accented versus unaccented beats, they obtained an average single-trial accuracy of 70% for perception and 61% for imagery. These results are very encouraging to further investigate the possibilities for retrieving information about the perceived rhythm from EEG recordings.

Very recently, the potential of deep learning techniques for neuroimaging has been demonstrated for functional and structural magnetic resonance imaging (MRI) data [15]. However, applications of deep learning techniques within neuroscience and specifically for processing EEG recordings have been very limited so far. Wulsin et al. [16] used deep belief nets (DBNs) to detect anomalies related to epilepsy in EEG recordings of 11 subjects by classifying individual "channel-seconds", i.e., one-second chunks from a single EEG channel without further information from other channels or about prior values. Their classifier was first pre-trained layer by layer as an autoencoder on unlabelled data, followed by a supervised fine-tuning with backpropagation on a much smaller labeled data set. They found that working on raw, unprocessed data (sampled at 256Hz) led to a classification accuracy comparable to hand-crafted features. Langkvist et al. [17] similarly employed DBNs combined with hidden Markov models (HMMs) to classify different sleep stages. Their data for 25 subjects comprised EEG as well as recordings of eye movements and skeletal muscle activity. Again, the data was segmented into one-second chunks. Here, a DBN on raw data showed a classification accuracy close to one using 28 selected features.

## 3 Data acquisition & pre-processing

### 3.1 Stimuli

The African rhythm stimuli were derived from recordings of traditional East African music [18]. The author (DC) composed the Western rhythmic stimuli. Rhythms were presented as sequences of sine tones that were 100ms in duration with intensity ramped up/down over the first/final 50ms and a pitch of either 375 or 500 Hz. All rhythms had a temporal structure of 12 equal units, in which each unit could contain a sound or not. For each rhythmic stimulus, two individual rhythmic sequences were overlaid whereby one sequence was played at the high pitch and the other at the low pitch. There were two groups of three individual rhythmic sequences for each cultural type of rhythm as shown in Table 1. With three combinations within each group and two possible pitch assignments, this resulted in six rhythmic stimuli for each group, 12 per rhythm type and 24 in total.[1] Finally, rhythmic stimuli could be played back at one of two tempi, having a minimum inter-onset interval of either 180 or 240ms.

Furthermore, we also formed groups based on how these stimuli were created. These allowed a more coarse classification with fewer classes. Ignoring the pitch assignments and thus considering the pairs [a,b] and [b,a] as equivalent, 12 groups were formed. At the next level, the stimuli derived from the same of the four groups of three sequences were grouped resulting in four groups of six stimuli. Finally, distinguishing East African from Western stimuli resulted in the binary classification problem that we addressed in our earlier work.

### 3.2 Study description

Sixteen East African participants were recruited in Kigali, Rwanda (3 female, mean age: 23 years, mean musical training: 3.4 years, mean dance training: 2.5 years). The participants first completed three behavioral tasks: a *rhythm discrimination task*, a *rhythm reproduction task*, and a *beat tapping task*. Afterward, thirteen subjects also participated in the EEG portion of the study. All participants were over

Table 1: Rhythmic sequences in groups of three that pairings were based on. All 'x's denote onsets. Larger, bold 'X's denote the beginning of a 12 unit cycle (downbeat).

| Western Rhythms | East African Rhythms |
|---|---|
| 1 **X** x x x  x x   x x    **X** x x x  x x   x x | 1 **X**  x x x x x   x x x x **X** x x x x x   x x x x |
| 2 **X**     x  x x    x   x **X**    x  x x     x   x | 2 **X** x  x   x    x   x **X**    x   x   x |
| 3 **X**   x x  x x   x x x x **X**  x x  x x   x x x x | 3 **X** x      x   x      **X** x   x   x |
| 4 **X**   x x  x x    x   x **X**  x x  x x      x   x | 4 **X** x x x   x x x   x x **X** x x x   x x x   x x |
| 5 **X** x x x   x x   x    **X** x x x   x x   x | 5 **X** x x  x x  x x  x **X** x x  x x  x x  x |
| 6 **X**   x x  x x   x x x x **X**  x x  x x   x x x x | 6 **X** x x  x x  x   x  **X** x x  x x   x   x |

the age of 18, had normal hearing, and had spent the majority of their lives in East Africa. They all gave informed consent prior to participating and were compensated for their participation, as per approval by the ethics boards at the Centre Hospitalier Universitaire de Kigali and the University of Western Ontario. The participants were instructed to sit with eyes closed and without moving for the duration of the EEG recording, and to maintain their attention on the auditory stimuli. All rhythms were repeated for 32 seconds, presented in counterbalanced blocks (all East African rhythms then all Western rhythms, or vice versa), and with randomized order within blocks. 12 rhythms of each type were presented – all at the same tempo, and each rhythm was preceded by 4 seconds of silence. EEG was recorded via a portable Grass EEG system using 14 channels at a sampling rate of 400Hz and impedances were kept below 10kΩ.

### 3.3 Data pre-processing

EEG recordings are usually very noisy. They contain artifacts caused by muscle activity such as eye blinking as well as possible drifts in the impedance of the individual electrodes over the course of a recording. Furthermore, the recording equipment is very sensitive and easily picks up interferences from the surroundings. For instance, in this experiment, the power supply dominated the frequency band around 50Hz. All these issues have led to the common practice to invest a lot of effort into pre-processing EEG data, often even manually rejecting single frames or channels. In contrast to this, we decided to put only little manual work into cleaning the data and just removed obviously bad channels, thus leaving the main work to the deep learning techniques. After bad channel removal, 12 channels remained for subjects 1–5 and 13 for subjects 6–13.

We followed the common practice in machine learning to partition the data into *training*, *validation* (or model selection) and *test* sets. To this end, we split each 32s-long trial recording into three non-overlapping pieces. The first $T$ seconds after an optional offset were used for the validation set. The rationale behind this was that we expected that the participants would need a few seconds in the beginning of each trial to get used to the new rhythm. Thus, the data would be less suited for training but might still be good enough to estimate the model accuracy on unseen data. The main part of each recording was used for training and the remaining $T$ seconds for testing. The time length $T$ was tempo-dependent and corresponded to the length of a single bar in the stimuli. Naturally, one would prefer segments that are as long as the 2-bar stimuli. However, this would have reduced the amount of data left for training significantly and since only the East African rhythm sequences 2 and 3 had differences between the first and second bar (cf. Table 1), we only used 1 bar. With the optional offset, the data sets were aligned to start at the same position within a bar.[2] The specific values for the two tempi are listed in Table 2. Furthermore, we decided to process and classify each EEG channel individually. Combining all 12 or 13 EEG channels in the analysis might allow to detect spatial patterns and most likely lead to an increase of the classification performance. However, this would increase the model complexity (number of parameters) by a factor of more than ten while at the same time reducing the number of training and test examples by the same factor. Under these conditions, the amount of data would not be sufficient to effectively train the CNN and lead to severe overfitting.

The data was finally converted into the input format required by the CNN to be learned.[3] If the network just took the raw EEG data, each waveform was normalized to a maximum amplitude of 1 and then split into equally sized frames of length $T$ matching the size of the network's input layer. No windowing

Table 2: Differences between slow and fast stimuli.

| tempo | participants | beat length | bar length $T$ | bars | optional offset | training segment length |
|-------|-------------|-------------|----------------|------|-----------------|-------------------------|
| fast | 1–3, 7–9 | 180ms | 2160ms | 14.815 | 1760ms | 27680ms - offset |
| slow | 4–6, 10–13 | 240ms | 2880ms | 11.111 | 320ms | 26240ms - offset |

function was applied and the hop size (controlling the overlap of consecutive windows) was either 24, which corresponded to 60ms at the sampling rate of 400Hz, or the equivalent of $T$ in samples. If the network was designed to process the frequency spectrum, the processing involved:

1. computing the short-time Fourier transform (STFT) with given window length of 96 samples and a hop size of 24 (This resulted in a new frequency spectrum vector every 60ms.),
2. computing the log amplitude,
3. scaling linearly to a maximum of 1 (per sequence),
4. (optionally) cutting of all frequency bins above the number requested by the network,
5. splitting the data into frames of length $T$ (matching the network's input dimensionality) with a given hop size of 1 (60ms) or the equivalent of $T$.

Hops of 60ms were chosen as this equals to one fourth or one third of the beat length in the slow and fast rhythms respectively. With this choice, we hoped to be able to pick up beat-related effects but also to have a window size big enough for a sufficient frequency resolution in the spectrum. Including the zero-frequency band, this resulted in 49 frequency bins up to 200Hz with a resolution of 4.17Hz. Using the log amplitude in combination with the normalization had turned out to be the best approach in our previous experiments trying to distinguish East African from Western stimuli [4].

## 4   Experiments

CNNs, as for instance described in [3], have a variety of structural parameters which need to be chosen carefully. In general, CNNs are artificial neural networks (ANNs) with one or more convolutional layers. In such layers, linear convolution operations are applied for local segments of the input followed by a non-linear transformation and a pooling operation over neighboring segments. If the EEG data is represented as waveform, the input has only one dimension (width) which corresponds to the time. If it is represented as frequency spectrum, it has a second dimension (height) which corresponds to the frequency. The kernel for each convolution operation is described by a weight matrix of a certain shape. Here, we only considered the *kernel width* as free parameter and kept the height maximal. Multiple kernels can be applied in parallel within the same layer whereby each corresponds to a different output *channel* of the layer. The *stride* parameter controls how much the kernels should advance on the input data between successive applications. Here, we fixed this parameter at 1 resulting in a maximal overlap of consecutive input segments. Finally, the *pooling* parameter controls how many values of neighboring segments are aggregated using the $max$ operation.

Like in our previous work, we used a DLSVM output layer as proposed in [19].[4] This special kind of output layer for classification uses the hinge loss as cost function and replaces the commonly applied softmax. The convolutional layers applied the rectifier non-linearity $f(x) = max(0,x)$ which does not saturate like sigmoid functions and thus facilitates faster learning as proposed in [20]. The input length in the time dimension was adapted to match the bar length $T$. All models were trained for 50 epochs using stochastic gradient descent (SGD) (on mini-batches of size 100) with exponential decay of the learning rate after each epoch and momentum. The best model was selected based on the accuracy on the validation set. Furthermore, we applied dropout regularization [21]. In total, this resulted in four learning parameters with value ranges derived from earlier experiments:

- the initial learning rate (between 0.001 and 0.01),
- the exponential learning rate decay per epoch (between 1.0 and 1.1),
- the initial momentum (between 0.0 and 0.5), and
- and the final momentum in the last epoch (between 0.0 and 0.99)

and three structural parameters for each convolutional layer

- the kernel width (between 1 and the input width for the layer),
- the number of channels (between 1 and 30), and

- the pooling width (between 1 and 10).

In our previous work, we successfully applied CNNs with two convolutional layers to classify the perceived rhythms into types (East African vs. Western) as well as to identify individual rhythms in a pilot experiment [4]. However, we were only able to test a small number of manually tuned structural configurations, leaving a considerable potential for further improvement. Here, we took a systematic approach for finding good structural and learning parameters for the CNNs. To this end, we applied a Bayesian optimization technique for hyper-parameter selection in machine learning algorithms, which has recently been described by Snoek et al. [22] and has been implemented in *Spearmint* library.[5] The basic idea is to treat the learning algorithm's generalization performance as a sample from a Gaussian process and select the next parameter configuration to test based on the *expected improvement*. The authors showed that this way, the number of experiment runs to minimize a given objective can be significantly reduced while surpassing the performance of parameters chosen by human experts. We implemented[6] our experiments using *Theano* [23] and *pylearn2* [24]. The computations were run on a dedicated 12-core workstation with two *Nvidia* graphics cards – a *Tesla C2075* and a *Quadro 2000*.

We followed the common practice to optimize the performance on the validation set. Because the 24 classes we would like to predict were perfectly balanced, we chose the *accuracy*, i.e., the percentage of correctly classified instances, as primary evaluation measure.[7] Furthermore, ranking the 24 classes by their corresponding network output values, we also computed the *precision at rank 3 (prec.@3)* and the *mean reciprocal rank (MRR)* – two commonly used information retrieval measures. The former corresponds to accuracy considering the top three classes in the ranking instead of just the first one. The latter is computed as:

$$MRR = \frac{1}{|D|} \sum_{i=1}^{|D|} \frac{1}{rank_i} \tag{1}$$

where $D$ is the set of test instances and $rank_i$ is the rank of the correct class for instance $i$. The value range is $(0,1]$ where the best value, 1, is obtained if the correct class is always ranked first.

## 4.1 Impact of pre-processing (subject 4)

At first, we analyzed the impact of the pre-processing on the performance of a model with a single convolutional layer. For this, we only considered the recordings from subject 4 who were easiest to classify in our earlier experiments. The exponential learning rate decay was fixed at 1.08 leaving three structural and three learning parameters for the Bayesian optimization. Results are shown in Figure 1 (left).

Generally, CNNs using the frequency spectrum representation were faster. A possible reason could be that the graphics cards performed better using two-dimensional kernels instead of long one-dimensional ones. Furthermore, the search for good parameters was much harder for the waveform representation because the value range for the kernel width was much wider ($[1,1152]$ instead of $[1,45]$). Thus, the search took much longer. For instance, using the large hop size, an accuracy of more than 20% was only achieved after 208 runs for CNNs using waveform input with offset and after 47 runs without offset. Comparable values were already obtained after 1 and 2 runs respectively for the CNNs with frequency spectrum input and the values shown in Figure 1 (left) were obtained after 45 and 105 runs respectively. Consequently, the frequency spectrum appeared to be the clearly preferable choice for the input representation.

With the small hop size of 60ms, a lot more training instances were generated because of the high overlap. This slowed down learning by a factor of more than 10. Hence, fewer configurations could be tested within the same time. Overall, the large hop size corresponding to 1 bar was favorable because of the significant speed-up without an impact on accuracy. By using the offset in combination with the hop size of 1 bar, all instances for training, validation and testing were aligned to the same position within a bar. This could explain the increase in accuracy for this parameter combination together with the spectrum representation. In combination with the waveform input, the inverse effect was observed. However, as it was generally harder to find good solutions in this setting, it could be that testing more configurations eventually would lead to the same result as for the spectrum.

|  | hop size | offset | waveform | | freq. spectrum | |
|---|---|---|---|---|---|---|
| **CNN** | 60ms | no | 33.3% | 233.7s | 33.7% | 22.3s |
| | (60 runs) | yes | 34.8% | 119.5s | 33.0% | 16.4s |
| | 1 bar | no | 33.0% | 12.7s | 33.3% | 0.4s |
| | (300 runs) | yes | 24.7% | 5.3s | **35.8%** | **0.3s** |
| **SVM** | 60ms | no | training did not finish | | | |
| | | yes | within 48 hours | | | |
| | 1 bar | no | 11.1% | | 22.2% | |
| | | yes | 12.2% | | 24.3% | |

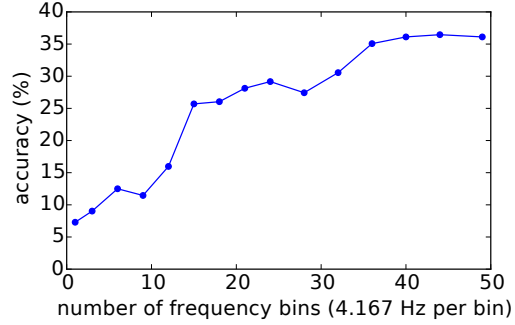

Figure 1: Impact of pre-processing. Left: Classification accuracy and average epoch processing time for different combinations of the pre-processing parameters. CNN structural and learning parameters were obtained through Bayesian optimization for 300 runs for hop size 1 bar and 60 runs for hop size 60ms. Processing times for CNNs were measured separately as single process using the *Tesla C2075* graphics card and averaged over 50 epochs. For comparison, SVM classification accuracies were obtained using *LIBSVM* with polynomial kernel (degree 1–5). (Only the best values are shown.) Right: Impact of the optional frequency bin cutoff on the accuracy.

For a comparison, we also trained support vector machine (SVM) classifiers using *LIBSVM* [25] on the same pre-processed data. Here, training did not finish within 48 hours for the small hop size because of the amount of training data. For waveform data, a polynomial kernel with degree 2 worked best, whereas for the frequency spectrum, it was a polynomial kernel with degree 4. All values were significantly (more than 10% absolute) below those obtained with a CNN. This shows using CNNs leads to a substantial improvement.

Next, we analyzed the impact of the optional frequency bin cutoff. To this end, we used the best pre-processing parameter combination from the above comparison. This time, we fixed the momentum parameters to an initial value of 0.5 and a final value of 0.99 as these clearly dominated within the best configurations found so far. Instead, we did not fix the exponential learning rate decay. This resulted in 5 parameters to be optimized. We sampled the number of frequency bins from the range of $[1,49]$ with higher density for lower values and let the Bayesian optimization run 300 experiments for each value. Results are shown in Figure 1 (right). A very significant accuracy increase can be observed between 12 and 15 bins which corresponds to a frequency band of 45.8–62.5 Hz in the high gamma range which has been associated with beat perception, e.g., in [5]. The accuracy increase between 28 and 36 bins (116–145 Hz) is hard to explain as EEG frequency ranges beyond 100 Hz have barely been studied so far. Here, a further investigation of the learned patterns (reflected in the CNN kernels) could lead to more insight. This analysis is still subject of ongoing research. The effect on the processing time was negligible.

Based on these findings, we chose the following pre-processing parameters for the remaining experiments: The EEG data was represented as frequency spectrum using 49 bins. Input frames were obtained with a hop size corresponding to the length of 1 bar, $T$, and with a offset to align all instances to the same position within a bar.

### 4.2 One vs. two convolutional layers (all subjects)

Having determined the optimal pre-processing parameters for subject 4 and CNNs with a single convolutional layer, we also used these settings to train individual models with one and two convolutional layers for all subjects. This time, we allowed 500 runs of the Bayesian optimization to find the best parameters in each setting. Additionally, we considered three groups of subjects. The 'fast' and 'slow' group contained all subjects with the respective stimulus tempo (cf. Table 2) whereas the 'all' group contained all 13 subjects. For the groups, we stopped the Bayesian optimization after 100 runs as there was no more improvement and the processing time was much longer due to the bigger size of the combined data sets. Results are shown in Table 3. Apart from the performance values for classifying individual instances that correspond a segment from an EEG channel, we also aggregated all predictions from the 12 or 13 different channels of the same trial into one prediction by a simple majority vote. The obtained accuracies are listed in Table 3 (right). Additionally, we computed the accuracies for the more coarse variants of the classification problem with fewer classes (cf. Section 3.1).

Table 3: Structural parameters and performance values of the best CNNs with one or two convolutional layers after Bayesian parameter optimization for each subject (500 runs) and the three subject groups (100 runs). Layer structure is written as [kernel shape] / pooling width x number of channels. (A more detailed table can be found in the supplementary material.)

| | | network structure | | channel mean (24 classes) | | | aggregated trial accuracy | | | |
|---|---|---|---|---|---|---|---|---|---|---|
| subject | input | 1st layer | 2nd layer | accuracy | prec.@3 | MRR | 24 classes | 12 classes | 4 classes | 2 classes |
| 1 | 33x49 | [5x49]/3x16 | [16x1]/5x12 | 19.1% | 36.1% | 0.34 | 25.0% | 29.2% | 58.3% | 79.2% |
| 2 | 33x49 | [10x49]/1x22 | | 27.1% | 46.5% | 0.42 | 37.5% | 37.5% | 50.0% | 87.5% |
| 3 | 33x49 | [17x49]/1x30 | | 21.9% | 38.2% | 0.36 | 20.8% | 25.0% | 45.8% | 66.7% |
| 4 | 45x49 | [35x49]/1x30 | | 36.1% | 63.5% | 0.55 | 50.0% | 62.5% | 75.0% | 83.3% |
| 5 | 45x49 | [40x49]/2x30 | | 18.1% | 34.7% | 0.33 | 16.7% | 25.0% | 41.7% | 70.8% |
| 6 | 45x49 | [26x49]/5x30 | [1x1]/10x30 | 29.5% | 48.1% | 0.45 | 37.5% | 41.7% | 54.2% | 75.0% |
| 7 | 33x49 | [15x49]/1x13 | | 23.1% | 43.9% | 0.40 | 33.3% | 45.8% | 54.2% | 66.7% |
| 8 | 33x49 | [5x49]/2x21 | [2x1]/2x24 | 24.0% | 44.2% | 0.41 | 41.7% | 41.7% | 58.3% | 91.7% |
| 9 | 33x49 | [13x49]/2x21 | [6x1]/4x30 | 21.8% | 33.7% | 0.36 | 25.0% | 29.2% | 58.3% | 91.7% |
| 10 | 45x49 | [7x49]/1x30 | | 26.6% | 51.0% | 0.44 | 33.3% | 33.3% | 45.8% | 66.7% |
| 11 | 45x49 | [27x49]/1x30 | | 26.6% | 55.1% | 0.45 | 33.3% | 37.5% | 41.7% | 75.0% |
| 12 | 45x49 | [5x49]/5x30 | [5x1]/10x30 | 32.1% | 60.9% | 0.51 | 29.2% | 33.3% | 54.2% | 83.3% |
| 13 | 45x49 | [18x49]/10x21 | [1x1]/6x30 | 20.2% | 37.2% | 0.36 | 25.0% | 29.2% | 50.0% | 70.8% |
| mean (1 convolutional layer) | | | | 24.4% | 46.4% | 0.41 | 30.8% | 36.5% | 51.6% | 74.7% |
| mean (2 convolutional layers) | | | | 24.4% | 44.2% | 0.40 | 29.5% | 34.0% | 52.2% | 77.2% |
| fast | 33x49 | [8x49]/1x22 | | 9.7% | 22.1% | 0.23 | 10.4% | 16.7% | 35.4% | 66.7% |
| | 33x49 | [1x49]/1x30 | [17x1]/1x30 | 9.5% | 21.6% | 0.23 | 11.8% | 19.4% | 38.9% | 67.4% |
| slow | 45x49 | [31x49]/1x30 | | 9.9% | 22.9% | 0.24 | 10.7% | 13.7% | 32.7% | 56.5% |
| | 45x49 | [1x49]/10x23 | [12x1]/5x27 | 9.1% | 24.3% | 0.24 | 10.1% | 13.1% | 31.5% | 58.9% |
| all | 33x49 | [1x49]/1x30 | | 7.3% | 19.0% | 0.21 | 7.7% | 12.2% | 29.2% | 57.1% |
| | 33x49 | [3x49]/9x22 | [5x1]/5x18 | 7.2% | 18.4% | 0.20 | 8.7% | 12.5% | 31.4% | 55.4% |

As expected, models learned for groups of participants did not perform very well. Furthermore, the classification accuracy varied a lot between subjects with the best accuracy (36.1% for subject 4) twice as high as the worst (18.1% for subject 5). This was most likely due to strong individual differences in the rhythm perception. But it might at least have been partly caused by the varying quality of the EEG recordings. For instance, the signal was much noisier than usual for subject 5. For most subjects, the aggregation per trial significantly increased the classification accuracy. Only in cases where the accuracy for individual channels was low, such as for subject 5, the aggregation did not yield an improvement.

Overall, the performance of the simpler models with a single convolutional layer was on par with the more complex ones – and often even better. One possible reason for this could be that the models with two convolutional layers had twice as many structural parameters and thus it was potentially harder to find good configurations. Furthermore, with more weights to be learned and thus more degrees of freedom to adapt, they were more prone to overfitting on this rather small data set. Figure 2 (left) visualizes the confusion between the different rhythms for subject 4 where the best overall accuracy was achieved.[8] Remarkably, only few of the East African rhythms were misclassified as Western (upper right quadrant) and vice versa (lower left). For the East African music, confusion was mostly amongst neighbors (i.e., similar rhythms; upper left quadrant) – especially rhythms based on sequences 2 and 3 that were the only ones that cannot be captured correctly in a window of 1 bar – whereas for the Western rhythms, there were patterns indicating a strong perceived similarity between rhythm sequences 1 and 4. The accuracies obtained for the classification tasks with fewer classes (cf. Table 3, right) paint a similar picture indicating strong stimulus similarity as the main reason for confusion. In the mean confusion matrix, this effect is far less pronounced. However, it can be observed in most of the confusion matrices for the individual subjects.

The results reported here still need to be taken with a grain of salt. Because of the study design, there is only one trial session (of 32 seconds) per stimulus for each subject. Thus, there is the chance that the neural networks learned to identify the individual trials and not the stimuli based on artifacts in the recordings that only occurred sporadically throughout the experiment. Or there could have been brain processes unrelated

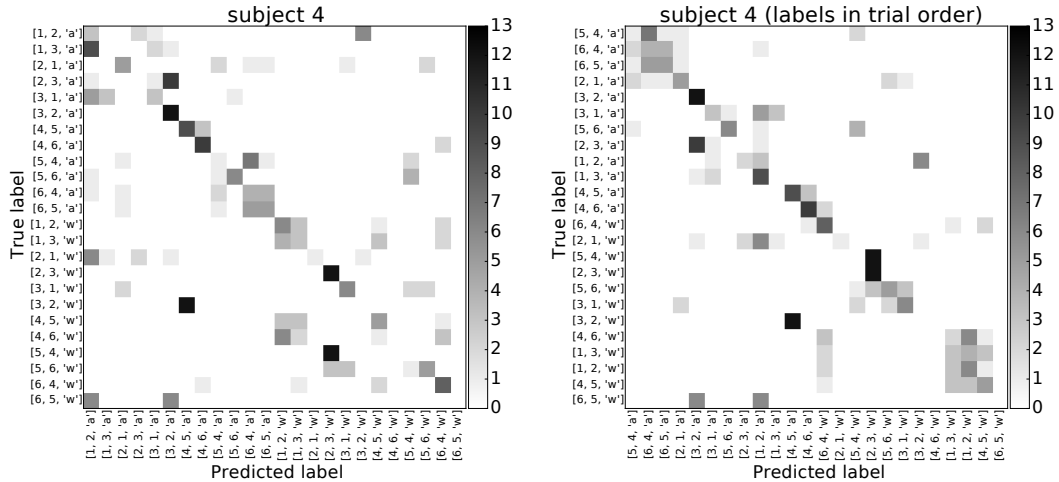

Figure 2: Confusion matrices for the CNN with a single convolutional layer for subject 4. Labels contain the ids of the high-pitched and low-pitch rhythm sequence (c.f. Table 1) and the rhythm type ('a' for African, 'w' for Western). Left: Labels arranged such that most similar rhythms are close together. Right: Labels in the order of the trials for this subject. More plots are provided in the supplementary material.

to rhythm perception that were only present during some of the trials. Re-arranging the labels within the confusion matrix such that they correspond to the order of the stimuli presentation (Figure 2, right) shows some confusion between successive trials (blocks along the diagonal) which supports this hypothesis. Repeating the experiment with multiple trials per stimulus for each subject should give more insights into this matter.

## 5    Conclusions

Distinguishing the rhythm stimuli used in this study is not easy as a listener. They are all presented in the same tempo and comprise two 12/8 bars. Consequently, none of the participants scored more than 83% in the behavioral rhythm discrimination test. Considering this and the rather sub-par data quality of the EEG recordings, the accuracies obtained for some of the participants are remarkable. They demonstrate that perceived rhythms may be identified from EEG recorded during their auditory presentation using convolutional neural networks that look only at a short segment of the signal from a single EEG channel (corresponding to the length of a single bar of a two-bar stimulus).

We hope that our finding will encourage the application of deep learning techniques for EEG analysis and stimulate more research in this direction. As a next step, we want to analyze the learned models as they might provide some insight into the important underlying patterns within the EEG signals and their corresponding neural processes. However, this is largely still an open problem. (As a first attempt, visualizations of the kernel weight matrices and of input patterns producing the highest activations can be found in the supplementary material.) We are also looking to correlate the classification performance values with the subjects' scores in the behavioral part of the study.

The study is currently being repeated with North America participants and we are curious to see whether we can replicate our findings. In particular, we hope to further improve the classification accuracy through higher data quality of the new EEG recordings. Furthermore, we want to conduct a behavioral study to obtain information about the perceived similarity between the stimuli. Finally, encouraged by our results, we want to extend our focus by also considering more complex and richer stimuli such as audio recordings of rhythms with realistic instrumentation instead of artificial sine tones.

### Acknowledgments

This work was supported by a fellowship within the Postdoc-Program of the German Academic Exchange Service (DAAD), by the Natural Sciences and Engineering Research Council of Canada (NSERC), through the Western International Research Award R4911A07, and by an AUCC Students for Development Award.

## Footnotes

[1]The 24 rhythm stimuli are available at http://dx.doi.org/10.6084/m9.figshare.1213903

[2] With offset, the validation and test set would correspond to the same section of the stimuli for the fast tempo whereas for the fast tempo, it would differ by 1 bar because of the odd number of bars in between.

[3] Most of the processing was implemented through the *librosa* library available at https://github.com/bmcfee/librosa/.

[4]We used the experimental implementation for *pylearn2* provided by Kyle Kastner at https://github.com/kastnerkyle/pylearn2/blob/svm_layer/pylearn2/models/mlp.py

[5] https://github.com/JasperSnoek/spearmint

[6] The code to run the experiments is available as supplementary material at http://dx.doi.org/10.6084/m9.figshare.1213903

[7] As the Bayesian optimization aims to minimize an objective, we let our learner report the *misclassification rate* instead which is one minus the accuracy.

[8]The respective confusion matrices for the models with two convolutional layers look very similar. They can be found in the supplementary material together with the matrices for the other participants.

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
