[Reviews · NeurIPS 2014]

Submitted by Assigned_Reviewer_41

This paper propose to use CNN to classify rhythms from EEG recordings.
A dataset with 13 subjects is analyzed.
Temporal and spatiotemporal (STFT) data representation are investigated.
The paper is well written with a good review of the relevant literature.
The experiments are well detailed in terms of parameters descriptions
and implementation used.

Results are relatively convincing. They show for example that spatiotemporal
representations shall be preferred, which is not surprising.

Some concerns and remarks:

- EEG channels are systematically processed individually whereas the physics
of the measurements state that channels are correlated due to
instantaneous mixing of the underlying neural sources. To me this
the strongest limitation of the method proposed, although is already
provides clearly above chance performance. This shall be discussed.

- The data analyzed are not public and no results with a baseline method
such as linear SVM are presented. Without making the data public or
providing such empirical results, it is hard to tell if CNN are
an overkill for the task.

- More discussion on Fig. 1 would be nice to understand what this
says in terms of neuroscience findings.
Summary: Well written, good literature review and relatively convincing results.

Submitted by Assigned_Reviewer_45

This paper addresses the problem of classifying rhythms based on EEG recordings using convolutional neural networks . This is an interesting application/study paper, and should be in the scope of this conference. However, there is no new algorithmic machine learning contribution.

- You mention that you didn’t do much pre-processing of the data with the hope that deep learning will figure things out. However, my understanding is that it is common practice to use ICA on EEG data as a standard pre-processing step. This could have cleaned up things quite a bit without a lot of effort from your side. It would be interesting to see if the results would have been better with this pre-processing.

- I suggest uploading the the stimuli used in the study on a webpage, and providing a link in the paper. Listening to the stimuli would help the reader have a better sense of the data.
Summary: This paper addresses the problem of classifying rhythms based on EEG recordings uses convolutional neural networks . This is an interesting application/study paper, but there is no new algorithmic machine learning contribution.

Submitted by Assigned_Reviewer_46

This paper uses Electroencephalography (EEG) recordings to study the possibility of determining the rhythm that a subject is listening to. The authors employ two different techniques: Stacked denoising autoencoders (SDA), and Convolutional Neural Network (CNN).

The authors have generated EEG data through experimentation. 13 subjects have listened to 24 different rhythms. Each EEG recording consists of about 13 channels. Each rhythm was 32 seconds long and EEG sampling frequency was 400Hz.

In one experiment, the authors train on the data from EEG signals of one individual and test on EEG data from the same individual. They use about 27 seconds of a 32 second rhythm for training and about 2.5 seconds for testing (exact time depends on rhythm tempo). They show that when trained and tested on a single individual, the accuracy of correctly classifying a rhythm into 24 classes is higher than chance (24.4% vs 4.17%). For one specific individual they can classify the EEG signal with 50.0% accuracy.

In another experiment they train using the data from all subjects. In this example the accuracy drops to about 10% (still 4.17% chance). In the case of binary classification (whether East African or Western rhythm) the accuracy of joint classification is 57.1% (where chance is 50%). This shows that it is difficult to have a classification model that is appropriate for all subjects. This phenomenon is understandable and is shown in voice recognition literature as well.

Although the dataset and the work are novel. There is little technical contribution in this paper. The paper primarily reports the results of two major algorithms on a new dataset. Although this is valuable on itself, I am not sure if it fits into the criteria of NIPS papers.
Summary: The paper is sound and demonstrates a full pipeline of data collection, experimentation and report. However, the technical contribution is limited as well-known techniques are tested on a dataset.
Author Feedback
Author rebuttal: We agree with the reviewers that our paper is an application/study paper with no new algorithmic machine learning contribution. However, we disagree with reviewer 46 on the point that its "technical contribution is limited."

To our knowledge, nobody has published results on using CNNs to classify EEG before. This application is not straightforward and there are many things to consider. Making bad design decisions can easily ruin the outcome. Hence, we took extra care to systematically test reasonable parameter configurations (e.g. for pre-processing) as well as to document technical details of the full pipeline and share the code for the experiments. This way, we hope to provide a valuable resource for others who would like to apply these techniques on similar data. We agree that we do not describe new techniques, but our technical contribution lies in describing _how_ existing techniques can successfully be applied in a new domain.

NIPS targets both, researchers who develop the kind of techniques applied here as well as those from the field of neuroscience who would benefit from applying them. The former might be interested in learning about a new promising application domain whereas the latter might want to know about our experiences for successfully applying CNNs themselves. This is why we think our paper is very relevant for NIPS. In our opinion, the audience could hardly be more suitable and we are hoping to get valuable feedback by presenting at NIPS.

In the following, we would like to address specific issues brought up by the reviewers. Many thanks for the constructive feedback! Including these points will clearly improve the paper.

- processing channels combined instead of individually (as in the paper):
Combining all 12 or 13 channels in the analysis would allow to detect spatial patterns and most likely lead to an increase of the classification performance. The CNN implementation already supports multiple channels (originally used for colours channels in images).
However, this would increase the model complexity (weight parameters) by a factor of more than 10 while at the same time reducing the number of training and test examples by the same factor. Under these conditions, the amount of data would not be sufficient to effectively train the CNN and lead to severe overfitting.

- possibility of using ICA for pre-processing:
ICA is commonly used to pre-process high-density EEG data. With only a few channels like in our dataset, its use becomes impractical - primarily because the number of detectable components is limited by the number of channels.

- comparing our results with (linear) SVM as a baseline
We re-computed the classification accuracy values from Table 3 (comparison of pre-processing settings) for a support vector machine classifier using libsvm with linear kernel and polynomial kernel (degree 2,3,4 and 5). The best obtained results were (in the same order as in table 3):
waveform: n/a n/a 11.1% 12.2%
spectrum: n/a n/a 22.2% 24.3%
(n/a - learning did not finish within 24 hours)
For waveform data, a polynomial kernel with degree 2 worked best, for the frequency spectrum, it was a polynomial kernel with degree 4. All values were significantly (more than 10% absolute) below those obtained with a CNN. This shows using CNNs leads to a substantial improvement.

- discussion of Fig. 1
The jump in classification accuracy for increasing the frequency cutoff from 45
to 62 Hz can be explained by prior findings in neuroscience. The 45-62 Hz band is in the high gamma range. Gamma activity has been associated with beat perception, e.g., in [1].

EEG frequency ranges beyond 100 Hz have barely been studied so far. Hence, it is hard to explain the increase between 116 and 145 Hz. Here, a further investigation of the learned patterns (reflected in the CNN kernels) could lead to more insight. This analysis is still subject of ongoing research.

[1] Snyder, Joel S., and Edward W. Large. "Gamma-band activity reflects the metric structure of rhythmic tone sequences." Cognitive brain research 24.1 (2005): 117-126.

- making the stimuli available
We have temporarily uploaded the stimuli to a dropbox folder:
https://www.dropbox.com/sh/6nlihp7b65sdoc1/AAAvWa0ii0XIuB2UfdTdcJY5a
The order and labels of the stimuli correspond to the labels in the confusion matrices.
For publication, we will include an official link to the stimuli together with the code.

- availability of the dataset
We regret that we are not allowed to share the dataset due to the ethics regulations of the study. We plan to collect new EEG recordings of music perception and imagination within this year for a new open access dataset and are currently applying for ethics approval.